# Long-distance propagation of short-wavelength spin waves

Chuanpu Liu[1,2], Jilei Chen[1], Tao Liu[3], Florian Heimbach[1], Haiming Yu [1], Yang Xiao[4], Junfeng Hu[1], Mengchao Liu[2], Houchen Chang[3], Tobias Stueckler[1], Sa Tu[1], Youguang Zhang[1], Yan Zhang[1], Peng Gao [2], Zhimin Liao [2], Dapeng Yu[2,5], Ke Xia[6], Na Lei[1], Weisheng Zhao[1] & Mingzhong Wu[3]

Recent years have witnessed a rapidly growing interest in exploring the use of spin waves for information transmission and computation toward establishing a spin-wave-based technology that is not only significantly more energy efficient than the CMOS technology, but may also cause a major departure from the von-Neumann architecture by enabling memory-in-logic and logic-in-memory architectures. A major bottleneck of advancing this technology is the excitation of spin waves with short wavelengths, which is a must because the wavelength dictates device scalability. Here, we report the discovery of an approach for the excitation of nm-wavelength spin waves. The demonstration uses ferromagnetic nanowires grown on a 20-nm-thick $Y_3Fe_5O_{12}$ film strip. The propagation of spin waves with a wavelength down to 50 nm over a distance of 60,000 nm is measured. The measurements yield a spin-wave group velocity as high as 2600 m s$^{-1}$, which is faster than both domain wall and skyrmion motions.

[1] Fert Beijing Research Institute, School of Electronic and Information Engineering, BDBC, Beihang University, 100191 Beijing, China. [2] State Key Laboratory for Mesoscopic Physics and Electron Microscopy Laboratory, School of Physics, Peking University, 100871 Beijing, China. [3] Department of Physics, Colorado State University, Fort Collins, CO 80523, USA. [4] Department of Applied Physics, Nanjing University of Aeronautics and Astronautics, 210016 Nanjing, China. [5] Department of Physics, Southern University of Science and Technology, 518055 Shenzhen, China. [6] Department of Physics, Beijing Normal University, 100875 Beijing, China. Chuanpu Liu, Jilei Chen, Tao Liu and Florian Heimbach contributed equally to this work. Correspondence and requests for materials should be addressed to H.Y. (email: haiming.yu@buaa.edu.cn) or to M.W. (email: mwu@colostate.edu)

Magnetic moments in a material can precess about their equilibrium direction and a collection of such precessions can propagate in the material and is usually referred to as a spin wave. In recent years, there has been a strong interest in taking the advantage of spin waves for data transfer and wave-based computation[1–10], and this interest is powered mainly by the high potential of a beyond-CMOS, energy-efficient new technology[11,12]. The properties of spin waves depend on the strength of the dipolar and exchange interactions in the materials. Generally speaking, exchange-dominated spin waves have nm-wavelength ($\lambda$) and travel faster than dipolar waves, so they are more desirable for data transfer and processing applications[13–15].

Several approaches are available to excite spin waves in magnetic thin films, but none can be used to efficiently produce exchange spin waves with the short $\lambda$ and the fast speed demanded by computing applications. The most conventional approach is to apply a spatially non-uniform microwave field through, for example, a coplanar waveguide (CPW)[16] or a nm-wide stripline[17,18]. Careful design has allowed the use of this approach to excite spin waves with $\lambda = 370$ nm[18], but it seems extremely challenging to go beyond this due to impedance mismatching. More recent approaches involve the use of spin-transfer-torque (STT) oscillators[19–21] and grating couplers[22,23]. For the STT oscillator, the major issue is the large threshold current density. Furthermore, while tuning the amplitude of spin waves, an unwanted change in the frequency also occurs. The grating couplers provoke multi-directional emission of short-$\lambda$ spin waves from magnetic nanodisks[22,23], but cannot convert microwave energy into propagating spin waves efficiently. Very recently, it has been demonstrated that spin waves with $\lambda = 125$ nm could be driven by the magnetic vortex states in two antiferromagnetically coupled layers[24]; and spin waves with $\lambda = 1600$ nm could be excited within magnetic domains in strain-coupled heterostructures[25]. In both cases, however, the spin waves are confined within small elements or structures and cannot propagate over long distances. Besides, it has been proposed that building ferromagnetic (FM)/piezoelectric bi-layered cells on a spin-wave bus and using voltage-controlled strains in the piezoelectric layer to induce magnetization switching in the FM layer can excite short-$\lambda$ spin waves in the bus[12,26–28]. This approach is very promising but has not been demonstrated experimentally yet.

This article reports an approach for short-$\lambda$ spin-wave excitations that makes use of magnetization precession in periodic FM nanowires to drive spin waves in a neighboring magnetic film. The experiments used Co nanowire arrays fabricated on a long and narrow $Y_3Fe_5O_{12}$ (YIG) thin-film strip with the wires transverse to the YIG strip. Upon the excitation of the precessional motion of the magnetic moments in the nanowires, the dynamical dipolar fields from the nanowires drive spin waves in the YIG thin film that have quantized wavenumbers associated with the period of the nanowire arrays. The shortest $\lambda$ detected experimentally is 50 nm; but it can be even shorter if finer wire arrays are used, as demonstrated numerically. Even though the spin waves are resonantly excited, they propagate through the YIG strip portion where there are no nanowires. Propagation over distances as long as 60,000 nm has been experimentally demonstrated. The group velocity has been measured, with the highest value reaching 2600 m per s. This velocity is much higher than the motions of domain walls (750 m per s)[29] and skyrmions (100 m per s)[30], both having been proposed as nanoscale information carriers.

## Results

**Nanopatterned magnetic heterostructure**. A schematic of a magnetic YIG/Ti/Co heterostructure is shown in Fig. 1a. The 20-nm-thick YIG thin film was grown on a gadolinium gallium garnet ($Gd_3Ga_5O_{12}$, GGG) substrate by magnetron sputtering. The Gilbert damping constant of the YIG film was found to be as low as $8 \times 10^{-5}$ (see Supplementary Note 1 for film characterization)[31]. The film was then patterned by ion beam etching to form a 90-μm-wide, spin-wave propagation channel. Periodic Ti(1 nm)/Co(25 nm) nanowires were fabricated on the top of the YIG channel using electron beam lithography (Methods). The nanowire array period $a$ stands for the center-to-center distance of two neighboring wires. The CPW on top is integrated to measure the spin-wave reflection spectra as shown in Fig. 1b–d. A transmission electron microscope (TEM) cross-section image of the YIG/Ti/Co heterostructure is shown in Fig. 1e. Its corresponding energy dispersive X-ray spectroscopy image for selected elements can be found in Supplementary Note 2. A bird's-eye-view scanning electron microscope (SEM) surface image of the heterostructure is shown in Fig. 1f. The external magnetic field ($H$) was applied parallel to the nanowires, and this field configuration corresponds to the presence of the Damon–Eschbach (DE) spin-wave modes in the YIG strip. A CPW structure was integrated on the top and its impedance was found to be ~50.4 Ohm, which was measured by the Smith chart on a vector network analyzer (VNA). Upon the application of a microwave signal to the CPW, the microwave magnetic field from the CPW excites the precessional motion of the magnetic moments in the Co nanowires and the YIG film. Although not shown, a second CPW was fabricated on the other end of the YIG strip and was used to detect the spin waves in the YIG. The scattering parameters of the device structure $S_{xy}$ ($x = 1,2$ and $y = 1,2$) were measured in both the reflection and transmission configurations using a VNA connected to the CPWs. $S_{21}$, for example, measures the spin-wave transmission from CPW antenna 1 to CPW antenna 2, as indicated in Fig. 2a. The magnetic field-dependent spin-wave reflection spectra $S_{11}$ are shown in Fig. 1b, d with opposite field-sweep directions. Different regimes are observed in the reflection spectra corresponding to the different magnetic configurations of the structure, as discussed below.

I.  Parallel configuration (P): The parallel alignment of the magnetization $\boldsymbol{M}$ in all Co wires as well as the YIG film can be simply achieved by applying a sufficiently large field, either positive or negative. Such a magnetic state remains unchanged until $H$ ramps close to zero. The highest and lowest frequency branches in Fig. 1b represent the CPW-excited precessional modes in the Co and the YIG, respectively. These modes have a relatively small wavevector ($k$)[32,33]. The Co mode has a fairly high frequency at $H = 0$. This can be explained by a large demagnetization field associated with the shape of the Co nanowires, which has been studied systematically by Ding et al.[34] Using Eq. (1) in ref. [34],

$$\omega = [(\omega_0 + (N_{zz} - N_{yy})\omega_M)(\omega_0 + (N_{xx} - N_{yy})\omega_M)]^{1/2},$$ we can fit the Co modes and derive an effective demagnetization factor $N_{xx} = 0.03$. The calculation based on closed form solutions and the Co nanowire dimensions yields $N_{xx} = 0.23$, which is significantly larger than the fitting value. This disagreement was also found by Ding et al.[34] and was explained as the influence of the effective dipole pinning of dynamic magnetization at the element edges.

II. Antiparallel configuration (AP): In Fig. 1b, a switching behavior is clearly observed when the field changes from negative to positive. The slopes of all the YIG modes change the sign, whereas the slope of the Co mode remains unchanged. This implies that the magnetization of the YIG film switches while that of the Co nanowires does not, resulting in an antiparallel configuration for the magnetization vectors in the YIG film and the Co nanowires. This

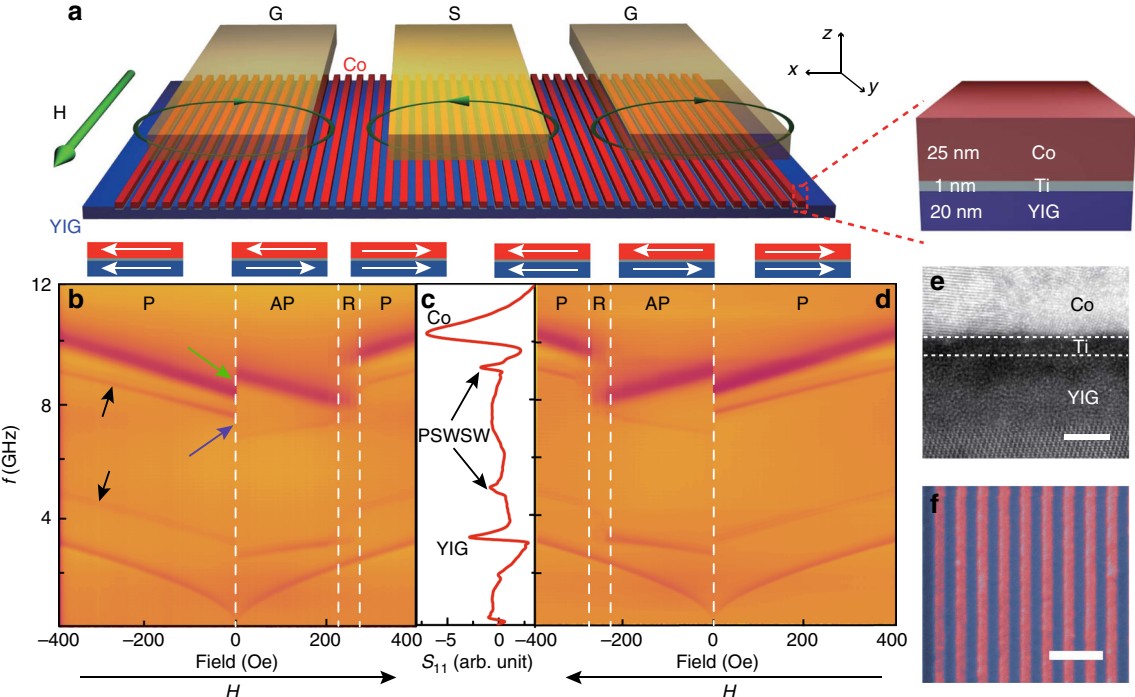

**Fig. 1** Different magnetic states in a nanopatterned magnetic heterostructure. **a** Sketch of an YIG (20 nm)/Ti (1 nm)/Co (25 nm) heterostructure with a coplanar waveguide (CPW) prepared on the top. The applied field $H$ is parallel to the Co nanowires (along the $y$ axis). **b** Color-coded reflection spectra $S_{11}$ measured on the device structure shown in **a**. The field is set to −3000 Oe to magnetize the Co nanowires and the YIG film to saturation first and then swept from −400 Oe to 400 Oe with a field step of 2.5 Oe. The spectra have several different regions corresponding to three different magnetic states: parallel state (P), antiparallel state (AP), and random state (R). **c** A line plot extracted from **b** at a field of 400 Oe. **d** Color-coded reflection spectra $S_{11}$ with a reversed field sweeping direction, as indicated. The field is set to 3000 Oe first and then swept from 400 Oe to −400 Oe with a field step of −2.5 Oe. **e** TEM image of the YIG/Ti/Co heterostructure. The horizontal scale bar is 2 nm long. **f** SEM surface image of the heterostructure. The Co nanowires are color-coded in red and the YIG film beneath the wires are in blue. The scale bar is 500 nm long

configuration remains unchanged until the external field is increased to about 230 Oe. Note that the antiparallel configuration exists mainly because the coercivity of the YIG film is smaller than that of the Co wires, thanks to the difference in the shape anisotropy. Note also that the formation of the antiparallel state should also be influenced by the interlayer magnetic couplings.

III. Random configuration (R): The third state is observed when the magnetic state changes from the AP back to the P. This transition regime is observed in a field range of about 60 Oe. It exists mainly because some of the Co wires switch earlier than the others when the field is increased gradually[35]. In this regime, some of the Co nanowires are in the P state while others in the AP state. Such a mixed state is further proved by minor-loop measurements where the resonances of the Co wires in both orientations coexist (Supplementary Note 3 and ref.[36]).

In all three configurations, we observe additional modes apart from the fundamental modes of the YIG and the Co. These modes, indicated by short black arrows in Fig. 1b, have similar characteristics as the YIG mode but resonant at much higher frequencies than the YIG mode. They are indeed high-wavenumber ($k$) spin-wave modes in the YIG film generated by the periodic modulation of the interlayer magnetic couplings with the Co nanowires. Interestingly, such high-$k$ spin waves can propagate over a remarkably long distance up to 60 μm (see Fig. 2 and Supplementary Note 4). In the following, we denote such modes as propagating short-wavelength spin waves (PSWSWs). More interestingly, unlike conventional standing spin waves due

to dimension confinement[37], the PSWSWs only have even number modes, e.g., $n = 2$, $n = 4$, and $n = 6$, due to periodically modulated boundary conditions where the magnetization precession is forced to stay in phase.

**Propagating short-wavelength spin waves**. Figure 2a shows a sketch of a measurement configuration for the demonstration of short-$\lambda$ spin-wave propagation. The propagation distance $s$ in such a device is about 30 μm. The Co nanowires are fabricated only beneath the CPWs, while the area in between the two CPWs remains bare YIG. Figure 2b shows transmission spectra $S_{21}$ for the $n = 4$ PSWSW, where a strong spin-wave transmission signal is detected. Such PSWSWs can propagate over a distance as long as 60 μm, as demonstrated using device A3 (see Table 1 and Supplementary Note 4). Such propagation distance is longer than recently discovered long-distance magnon transport[38]. Note that in ref. [38], the spin waves excited by a Pt strip are incoherent and therefore cannot be used for computing that utilizes spin-wave phase, whereas the spin waves described here are coherent, as demonstrated by the clear phase oscillation in Fig. 2b for instance. By extracting $\Delta f$ from the phase oscillations in Fig. 2c, we can determine the group velocity[33] of the spin waves as

$$v_{\mathrm{g}} = \frac{\mathrm{d}\omega}{\mathrm{d}k} \approx \frac{2\pi\Delta f}{2\pi/s} = \Delta f \times s. \tag{1}$$

Group velocities are extracted for all three PSWSW modes observed in the experiments, i.e., $n = 2, 4, 6$, together with the CPW-excited spin-wave mode (Fig. 2d). The group velocity for

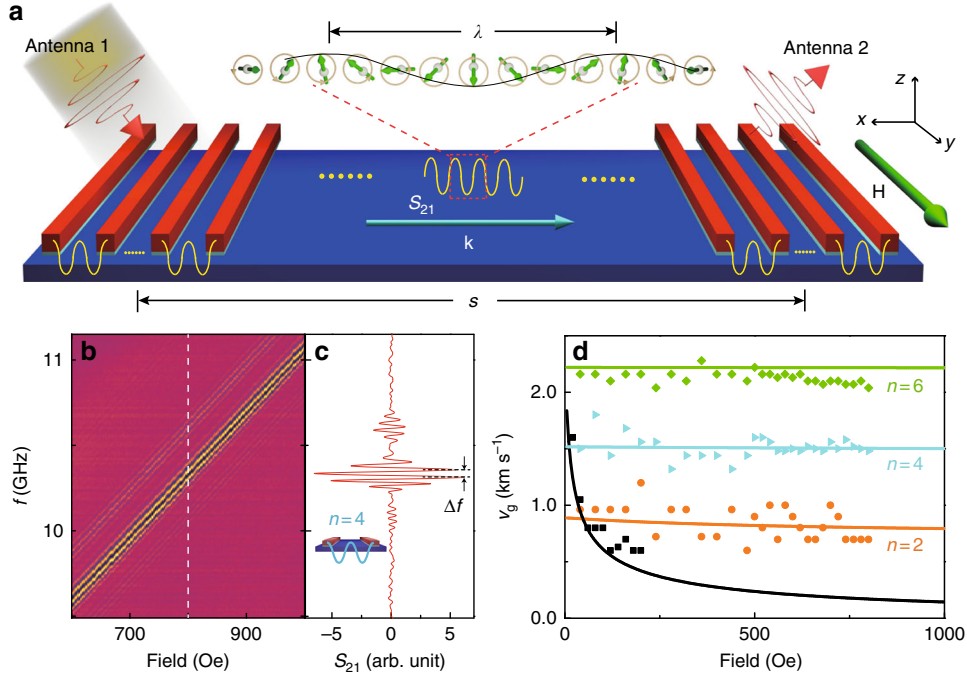

**Fig. 2** Propagating short-wavelength spin waves and their group velocities. **a** Sketch of a YIG/Ti/Co heterostructure-based device for spin-wave propagation measurements. The external field $H$ is in the plane and parallel to the Co nanowires. **b** Color-coded transmission spectra $S_{21}$ measured on a YIG/Ti/Co device with a nanowire period of 200 nm (device A1). The field is set to −3000 Oe first and then swept from 600 Oe to 1000 Oe with a field step of 2.5 Oe. **c** A line plot extracted as a cutoff from **b** at the field value of 800 Oe. $\Delta f$ is extracted for the calculation of the group velocity. **d** Group velocities of different spin-wave modes. Black squares show the group velocities of CPW-excited spin waves in the 20-nm-thick plain YIG film extracted from the experimental data on a YIG/Ti/Co structure with a nanowire period of 180 nm (device A2). Yellow circles, blue triangles, and green diamonds show the PSWSW group velocities extracted from the experimental data with mode number $n = 2$, $n = 4$, and $n = 6$, respectively. The solid lines are the group velocities calculated based on the derivative of the spin-wave dispersion relation in the YIG film

### Table 1 Parameters and properties of seven devices

| Device | Structure | Thickness (nm) | $a$ (nm) | $n$ | $\lambda$ (nm) | $s$ (μm) | $\eta$ |
|---|---|---|---|---|---|---|---|
| A1 | YIG/Ti/Co | 20/1/25 | 200 | 2, 4 | 100 | 30 | 95% |
| A2 | YIG/Ti/Co | 20/1/25 | 180 | 2, 4, 6 | 60 | 30 | 106% |
| A3 | YIG/Ti/Co | 20/1/50 | 180 | 4 | 90 | 60 | 61% |
| A4 | YIG/Al$_2$O$_3$/Co | 20/7/30 | 600 | 14, 16 | 75 | 30 | 11%[a] |
| B1 | YIG/Ti/Ni | 20/2/20 | 600 | 4, 6, 8, 10 | 120 | 30 | 21% |
| C1 | YIG/Ti/CoFe | 20/1/50 | 200 | 2, 4, 6, 8 | 50 | 15 | 98% |
| C2 | YIG/Al$_2$O$_3$/CoFe | 20/25/50 | 200 | 2, 4 | 100 | 15 | 41% |

Notes: $a$ is the nanowire array period; $n$ is the index of the observed spin-wave mode; $\lambda$ is the shortest spin-wave wavelength measured; and $s$ is the distance between the two CPWs. $\eta$ denotes the detected highest amplitude of the $n = 4$ spin-wave normalized by that of CPW-excited spin waves for an external field of 100 Oe
[a]This value is extracted for $n = 14$ mode, since the $n = 4$ mode is barely detectable for this sample

the $n = 6$ PSWSW mode reaches about 2200 m per s, which is higher than the group velocity of the CPW-excited spin waves over the full field range. In the dispersion shown in Fig. 3a, the derivative of the curve increases with $k$, so we can expect even higher group velocities for higher $k$ modes. The strong transmission signal indicates highly efficient excitation and detection of the PSWSWs. In ref. [23], the highest excitation efficiency achieved for the high-$k$ mode is about 30% of the efficiency for the CPW-excited mode, whereas in this work, the amplitude for the $n = 4$ mode is >100 times larger than the CPW-excited $k_1$ mode at the same field. At very low fields, e.g., 100 Oe, the amplitude for the $n = 4$ mode is still comparable with the $k_1$ amplitude (see Supplementary Note 5 and Table 1). If we take into account the influence of spin-wave attenuation during the

propagation[32], the estimated excitation efficiency is even slightly higher (supplementary information).

**Exchange spin-wave dispersion relations.** We extracted the frequencies for different PSWSW modes from the experiments on three devices (A1, A2, and C1), which are presented in Fig. 3. The corresponding $k$ values of the PSWSWs are determined by $k_{Sn} = \frac{n\pi}{a}$. We find it follows reasonably well with the dispersion relation calculated using[39]

$$\omega = \left[ \left( \omega_0 + \omega_M \lambda_{ex} k^2 \right) \left( \omega_0 + \omega_M \left( \lambda_{ex} k^2 + 1 \right) \right) \right]^{1/2}, \quad (2)$$

where $\omega = 2\pi f$, $\omega_0 = \gamma H$, $\omega_M = \gamma \mu_0 M_S$, and $\lambda_{ex}$ is the exchange constant. $\gamma$ stands for the absolute gyromagnetic ratio. $H$ is the

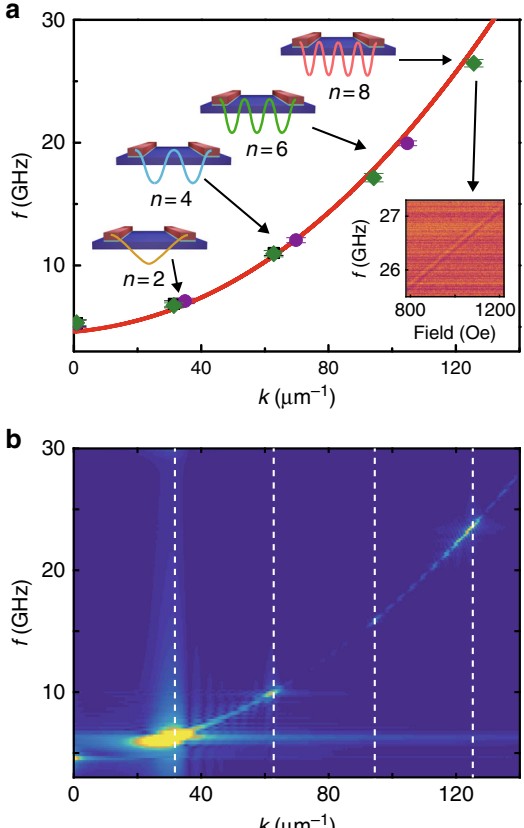

**Fig. 3** Dispersion relations for propagating short-wavelength spin waves. **a** Data points show the frequencies and wavenumbers of different spin-wave modes extracted from the experiments on device A1—YIG/Ti/Co with $a =$ 200 nm (black squares), device A2—YIG/Ti/Co with $a =$ 180 nm (purple dots), and device C1—YIG/Ti/CoFe with $a =$ 200 nm (green diamonds). The red curve shows the dispersion relation of the DE spin wave in the YIG thin film, which is calculated using Eq. (2) for a field of 1000 Oe. The inset shows the transmission spectra $S_{21}$ for the $n =$ 8 PSWSW mode detected in device C1. **b** Micromagnetic simulation results of the dispersion relation for the YIG/Ti/Co structure with a period of $a =$ 200 nm. The simulation takes into account the interlayer dipolar interactions between the Co and the YIG, but not direct interlayer exchange coupling

skyrmion motion. Using the experimental data in Fig. 3a and $v_p = 2\pi f/k$, one can further estimate the phase velocity to be 1320 m per s This value is about half of the group velocity, which provides additional evidence for the facts that the dispersion relation of the PSWSWs follows $f \propto k^2$ and the PSWSWs are exchange spin waves.

It is likely that the excitation of the PSWSWs relies on the periodic modulation of the effective field on the YIG film by the Co nanowires, through either the interlayer dipolar interaction or the interlayer exchange coupling. In order to further investigate this, we run full micromagnetic simulations on the YIG/Ti/Co heterostructures. Figure 3b shows the simulation results when we consider only the interlayer dipolar interaction and leave out the interlayer exchange coupling. Apparently, with only the dipolar interaction, we can already very well resolve the higher order modes from $n =$ 2 to $n =$ 8 (see Methods for more details). These results indicate that the interlayer dipolar interaction is the dominate mechanism responsible for the excitation of the PSWSWs. The interlayer dipolar interaction is also revealed by vibrating sample magnetometer measurements on multi-layered YIG/Ti/Co structures, bare YIG thin films, and bare Co thin films (Supplementary Note 7). Note that interlayer coupling in $Ni_{80}Fe_{20}/Ru/Ni_{80}Fe_{20}$ multi-layered films[40] and nanowire arrays[41] has been studied using FM resonance previously, and the results show that apart from the dipolar interaction, the interlayer exchange interaction also occurs but critically depends on the thickness of the spacer between the two magnetic layers.

In Fig. 1b, d, we can see that when the device switches from the P state to the AP state near $H =$ 0, the Co mode shifts up in frequency, as indicated by the blue arrow, by about 0.63 GHz. For the fundamental mode in the YIG, we see a frequency blueshift, although it is only about 0.08 GHz. These frequency shifts can be understood as a result of the interlayer dipolar interaction, since the effective interlayer dipolar field on the Co and the YIG reverses the direction when the magnetic state changes from the P configuration to the AP configuration. The observation that the frequency shift of the Co mode is notably larger than that of the YIG mode can be understood if we take the Kittle equation and consider the fact that the $M_S$ value of Co is >10 times larger than that of YIG[37].

For the PSWSW modes, however, the frequency shift (about 0.73 GHz) is much larger than that of the fundamental mode in the YIG (0.08 GHz). This result cannot be simply explained in terms of the interlayer dipolar interaction. It may be explained if we consider that in Eq. (2)[39], the terms proportional to $k^2$ are slightly influenced by the switching of the magnetic states between the P and the AP, since for high-$k$ PSWSWs such terms dominate over other terms. Such consideration is consistent with the observation that the $n =$ 4 mode (with a higher $k$) shows a larger redshift than the $n =$ 2 mode (with a smaller $k$). Note that $\lambda_{ex}$ in Eq. (2) stands for the exchange constant in the YIG and is determined by the material itself. Therefore, we assume an additional effective $\lambda_{ex}'$ that is induced by the interlayer exchange coupling in the structure.

To further examine the roles of the interlayer exchange coupling, we run micromagnetic simulations for both the P and AP states with both the interlayer dipolar and exchange interactions considered. Interestingly, with the interlayer exchange coupling as small as only 10% of the exchange constant[42], a shift in the PSWSW frequency is observed (Supplementary Note 8). If we tune the interlayer exchange coupling between FM-like and antiferromagnetic (AFM)-like, a reversed sign of the frequency shift is observed. If we consider a weak FM-like interlayer exchange coupling on top of the interlayer dipolar interaction, the experimentally observed red-shift in the PSWSW frequency can be well understood. Thus,

external field. When $k$ is large, the exchange term $\lambda_{ex}k^2$ dominates, and the dispersion follows approximately $f \propto k^2$ dependence. This dependence is indeed observed in the experimental results shown in Fig. 3a, demonstrating that the PSWSWs are exchange-dominated spin waves. In the calculation, we take $\lambda_{ex} = 3 \times 10^{-16}$ for the bare YIG film. The highest wavenumber detected experimentally is about 126 $\mu m^{-1}$ for device C1 (Table 1), whose transmission signal is shown in the inset of Fig. 3a. This mode extends over a fairly broad field range; and even at zero field, a strong transmission signal can still be measured (Supplementary Note 6). Such high-$k$ spin waves are already beyond the detection limit (19 $\mu m^{-1}$) of micro-focused Brillouin light scattering (BLS)[20], which is a technique widely used for spin-wave studies. This means that spin waves with wavelengths below 300 nm cannot be detected by the BLS technique. The corresponding wavelength of the $k =$ 126 $\mu m^{-1}$ mode is 50 nm, which is the shortest wavelength achieved so far for propagating spin waves to the best of our knowledge. Using Eq. (1), the group velocity of this mode is estimated to be 2600 m per s, which is higher than the velocity of typical domain wall and

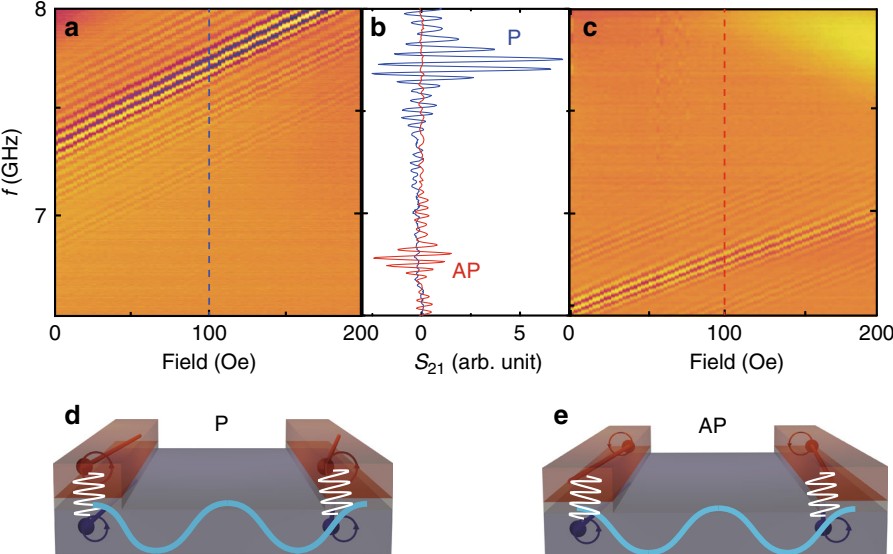

**Fig. 4** Reconfigurable short-wavelength spin waves. **a** Color-coded plot of transmission spectra $S_{21}$ measured in the $P$ state. The field is set to 3000 Oe first and then swept from 200 Oe to 0 with a field step of −2.5 Oe. **c** Color-coded transmission spectra $S_{21}$ measured in the AP state. The field is set to −3000 Oe first and then swept from 0 to 200 Oe with a field step of 2.5 Oe. **b** Line spectra taken as cuts from the full spectra at an applied field of 100 Oe. These cuts are indicated by the dashed lines in **a** and **c**. **d** and **e** illustrate the excitation of $n = 4$ short-wavelength spin waves induced by the interlayer magnetic coupling in the $P$ and AP configurations, respectively

based on the comparison of our experimental and numerical results, we can conclude that the interlayer dipolar interaction is the dominating mechanism for the excitation of the PSWSWs; and a weak FM-like interlayer exchange coupling also presents in the structure and makes notable influences on the excitation of high-$k$ modes. To verify this conclusion experimentally, we designed and fabricated device C1 and device C2, aiming at facilitating direct comparisons. The two devices are almost the same except that in device C2 the interlayer Ti is replaced by a 25-nm-thick $Al_2O_3$ layer so that the interlayer exchange coupling is excluded but the dipolar interaction is still present. In the experiments using device C2, we still see the $n = 2$ and $n = 4$ modes, but we no longer observe the frequency redshift when we switch between the $P$ and AP states. In contrast, the redshift is clearly observed in the measurements using device C1 (Supplementary Note 9). The same results have been observed for device A4 where a 7-nm-thick $Al_2O_3$ layer is inserted, although both transmission and reflection signals are much weaker because the $Al_2O_3$ layer is thick (Supplementary Note 10). These results evidently support our conclusion that the dipolar interaction is the major mechanism responsible for the PSWSW excitation, while the interlayer exchange coupling is responsible for the frequency shift of the PSWSWs.

**Reconfigurable spin-wave propagation.** As shown in Fig. 1b, d, with the same external field, e.g., 100 Oe, different magnetic configurations can be formed with different field sweeps. Thus, different PSWSWs can be excited for a same field by tuning the device structure between the $P$ and AP configurations. Figure 4a–c show the transmission spectra $S_{21}$ for the $n = 4$ PSWSW mode under the $P$ and AP configurations. We can see that for the same field, the frequency is shifted by 0.95 GHz and the transmission intensity is changed dramatically. The transmission signal in the AP state is only 40% of that in the $P$ state. This is to some extent analogy to the giant magnetoresistance (GMR)[43,44] in metallic spin valves. If we consider the spin-wave transmission signal as a spin-wave spin current[45], a magnonic

GMR ratio can be estimated as $\frac{S_{21}^P - S_{21}^{AP}}{S_{21}^P} \approx 60\%$. Such a critical dependence of the transmission signal on the magnetic configuration might be due to three possible reasons. (1) The magnetic moments in the YIG precess in the same manner as that in the Co for the $P$ state but in an opposite manner in the AP state, resulting in a different efficiency for the Co-drive-YIG scenario. (2) The spin current generated by the Co magnetization precession exerts anti-damping-like and damping-like torques[46,47] on the YIG magnetization for the $P$ and AP states, respectively, giving rise to a larger amplitude for the $P$ state than the AP state. (3) It is possible that the interlayer exchange coupling being FM-like helps the PSWSW excitation in the $P$ state. Note that some additional weaker spin-wave modes are also observed around the main modes in both the color-coded spectra and the spectrum cuts. These modes may be excited due to the CPW-associated grating coupler effect[22]. Fig. 4d, e are the sketches of the phase distribution of the PSWSWs in the $P$ and AP states, which show clearly the relation between the wavelength and the nanowire array period.

**Device comparisons: material dependence and scalability.** Turn now to the effects of the material of the nanowire array. Specifically, we change the nanowires from Co to Ni first and then CoFe. Table 1 summarizes the parameters and properties of seven different devices that we examined. The last column $\eta$ indicates the PSWSW efficiency compared with the efficiency of the excitation of long-$\lambda$ spin waves using the conventional CPW structures. In device B1, we have a Ni nanowire array with a period of $a = 600$ nm, and we observed PSWSWs with an index up to $n = 10$. Note that the $n = 10$ mode has a wavenumber of 52.4 $\mu m^{-1}$, which corresponds to a wavelength of 120. Note also that the PSWSW amplitude in this device is much lower than in device A1. Therefore, for Ni-based structures, it may not be a good idea to excite spin waves with shorter $\lambda$ values by reducing the array period, as the spin-wave amplitude would be even weaker. The main reason for this limitation is that Ni has a smaller $M_S$ than Co, therefore a weaker interlayer dipolar interaction. This is

consistent with the conclusion drawn from the micromagnetic simulations that the dominating mechanism of the PSWSW excitation is the interlayer dipolar interaction. More discussions on why some modes are not detectable are given in the Supplementary Information.

The PSWSW efficiency is found to be enhanced when its frequency approaches the resonance of the nanowires. This is likely because, for a given microwave power, the angle of the magnetization precession in the nanowires is larger when the precession frequency is closer to the ferromagnetic resonance (FMR) frequency, giving rise to higher dynamical dipolar fields on the magnetization in the YIG in a manner similar to the resonant transducer effect reported previously[48].

From the above discussions, we can see that it is desirable to replace Co with a material exhibiting a higher $M_S$ and meanwhile showing reasonably low damping. As studied by Schoen et al.[49], CoFe is one ideal candidate for such a purpose. In device C1 where the Co wires have been replaced by a 50-nm-thick CoFe nanowires, the spin-wave mode with the highest $k$ is excited, which corresponds to a PSWSW with the shortest wavelength observed, being 50 nm (inset of Fig. 3a). In order to achieve an even smaller wavelength, we can use finer nanowire arrays, e.g., with $a = 20$ nm. Simulation results show that PSWSWs with $\lambda \sim 10$ nm and $v_g \sim 13.1$ km per s can be achieved at a frequency of about 466 GHz entering already sub-THz regime (Supplementary Note 11). Spin waves with such properties are highly desirable in terms of the scalability and speed of spin-wave-based computing.

## Discussion

We have demonstrated the use of FM nanowires on the top of a magnetic thin film to efficiently excite exchange-dominated, high-speed, short-wavelength spin waves in the thin film. Spin waves with a wavelength as short as 50 nm propagating over a distance as long as 60,000 nm at a speed of 2600 m per s are measured experimentally and simulated numerically. Remarkably, the excitation and detection efficiency of such sub-100 nm-wavelength exchange spin waves is found to be as high as the excitation of dipolar spin waves using the conventional CPW-based techniques. The comparisons between the experimental results on different hybrid nanostructures suggest that the material engineering of such devices is vital and may significantly enhance the functionalities and capabilities. One promising further step will be using materials such as barium hexagonal ferrites that show a resonance frequency as high as 60 GHz and a damping constant much lower than that in FM metals[50]. The micromagnetic simulations suggest that by reducing the nanowire array period, it is possible to excite spin waves that have even smaller wavelengths and higher velocities than those demonstrated experimentally in this work. Ultimately, ultra-high-speed spin-wave propagation is envisaged based on AFM devices[51,52].

## Methods

**Device fabrication**. The 20-nm-thick YIG film was grown on a 0.5-mm-thick (111)-oriented gadolinium gallium garnet substrate by sputtering. By using optical lithography and ion beam etching, a YIG waveguide (width 90 μm, length 250 μm) was created. To avoid interference with reflected spin waves, the waveguide is tapered at both ends. Magnetic nanowires were fabricated on the YIG waveguide using electron beam lithography and lift-off process. The stacking-structured nanowires were prepared by electron beam evaporation. The titanium is used as an adhesion layer and the gold protects the cobalt (Ni or CoFe) from oxidation. Two CPWs (signal and ground line width 2 μm, edge-to-edge separation 1.6 μm) were patterned on top of the nanowires in the parallel orientation. The spatial $k$ distribution of the dynamic field can be calculated by using the dimensions of the CPW and a Fourier transform. The distance $s$ between two CPWs varies from 15 to 60 μm.

**Spin-wave measurements**. All measurements were carried out with a VNA-based all electrical spin-wave spectroscopy. The frequency range covered 10 MHz–40 GHz. By connecting CPWs via microwave probes to the VNA, an oscillating radio frequency (RF) field was generated, which leads to the excitation of spin waves. The magnetic flux changes caused by the spin waves induced an RF current in the antennas and was detected by the VNA. The absorption spectra in reflection and transmission configuration was measured from the $S$ parameters of the VNA. The power used in the experiment was 0 dBm. The external magnetic field $H$ was applied perpendicular to the YIG waveguide so the DE spin waves were generated.

**Micromagnetic simulations**. The program OOMMF (http://math.nist.gov/oommf) was utilized. The simulated structure consists of a YIG waveguide (30 μm × 5 μm × 20 μm) ($x\ y\ z$) and 9 cobalt stripes (length ($y$) 5 μm, thickness ($z$) 30 nm) placed on top in the center of the YIG. If not mentioned otherwise, we used an $M_S = 139.26$ kAm$^{-1}$, an exchange coefficient $A = 3 \times 10^{-12}$ J m$^{-1}$, a damping constant $\alpha = 2.3 \times 10^{-4}$ for YIG and $M_S = 1350$ kAm$^{-1}$, $A = 13 \times 10^{-12}$ J m$^{-1}$, and $\alpha = 0.01$ for the cobalt stripes. The mesh was set to 5 nm × 500 nm × 10 nm. To excite the magnetization, an excitation area was set around the stripes. Within this area, we had a uniform field in $x$ direction changing with time $t$ and $H_0 = 10$ mT according to $H_{ex} = H_0 \sin(2\pi \times 100\,\text{GHz}(t - 100.1\,\text{ps}))/(2\pi \times 100\,\text{GHz}(t - 100.1\,\text{ps}))$. An additional global bias field was applied in the $y$ direction. The ground state of the magnetization was determined by minimizing the energy of the structure. Starting from there the time-dependent magnetization has been simulated for 1000 equidistant times in steps of 5 ps. In the evaluation, the acquired magnetic configurations have been used to perform a two-dimensional fast fourier transformation (FFT) in time and $x$ direction. For the results shown, this method has been performed on the cells on one side of YIG, not covered by stripes. Therefore, the dispersion relations represent excited spin waves propagating from the center (excitation area with stripes) into the unpatterned YIG film.

**Data availability**. The data that support the findings of this study are available from the corresponding author upon request.

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

## Acknowledgements

We thank G. Bauer, C.E. Patton, K. Buchanan, and M.B. Jungfleisch for helpful discussions and S. Granville for improving the manuscript. We wish to acknowledge the support by NSF China under grant nos. 11674020, 11444005, youth 1000 talent program, 111 talent program B16001, and Ministry of Science and Technology of China MOST no. 2016YFA0300802. The work at CSU was mainly supported by the SHINES, an Energy Frontier Research Center funded by the US Department of Energy, Office of Science, Basic Energy Sciences under award SC0012670. The characterization of YIG thin films was supported by the US National Science Foundation under award EFMA-1641989.

## Author contributions

C.L. and H.Y. conceived and designed the experiments. T.L., H.C. and M.W. provided the high-quality YIG films and characterized them using FMR measurements. C.L., J.C., J.H., Z.L., D.Y. and H.Y. designed and fabricated the spin-wave devices. C.L., M.L. and P.G. conducted the TEM, SEM, and energy dispersive X-ray spectroscopy (EDX) characterization. J.C., F.H., T.S., S.T., Y.Z., W. Z. and H.Y. contributed to the experimental setup. J.C., F.H., Ya.Z. and H.Y. optimized the integrated microwave antenna. J.C., C.L., T.S. and H.Y. performed the measurements. C.L., J.C., T.L., N.L., H.Y. and M.W. analyzed the data. F.H., Y.X. and K. X. performed the theoretical simulations and analysis. H.Y. and M.W. supervised the experimental study. H.Y., J.C., C.L. and M.W. wrote the paper and the supplementary information with help from all the other co-authors. All authors commented on the manuscript.

## Additional information

**Competing interests:** The authors declare no competing financial interests.

