## [Peer Review File · Nature Communications]

Reviewers' comments:

Reviewer #1 (Remarks to the Author):

The authors describe measurements on nanowires that yield short-wavelength spin waves and fast propagation velocities. These are key ingredients for any potential future implementation of spin-wave based memory-logic technology. The manuscript presents a systematic study of different thickness and structure dependence and offer routes to lowering the shortest wavelength.

The shortest wavelength is reported as 50nm in the abstract. However this is presented as an estimate in the main body of the text since the sensitivity of the measurement is not sufficient. This therefore prompts the question of whether this is an incremental advance or something more distinct. For example there are reports of wavelengths of around 100nm following a different method recently (Phys. Rev. Applied 8, 014020 (2017)).

Therefore the claim of shortest wavelength "ever achieved" should be toned down. Instead the importance of this manuscript is the new methodology that allows shorter wavelength and long propagation and whether this is a suitably sufficient advance. The standard dipolar exchange is certainly involved, however the authors claim that including exchange interactions is necessary to explain the behavior and this is the key to the shorter wavelength and faster group velocity. The manuscript presents reasoned arguments and data to support this claim and is sufficiently novel and of sufficient interest.

A further comment:

In Fig 2.d the solid lines match $n=2$ and $n=4$ group velocities, however at $n=6$ there is some departure at higher field. Is this an indication of a break down of the model for higher n /field that may inhibit the path to shorter wavelengths/faster group velocity.

Reviewer #2 (Remarks to the Author):

The manuscript "Long-distance propagation of short-wavelength spin waves" by C. Liu et al. represents interesting experimental studies of excitation and propagation of exchange spin waves

in thin YIG films. The Authors present interesting results, nevertheless I can not recommend the manuscript for the publication in Nature Communications unfortunately. It is stated in the manuscript “This article reports a new approach for short- λ spin wave excitations that makes use of magnetization precession in periodic ferromagnetic nanowires to drive spin waves in a neighboring magnetic film.” I can not agree with this statement. The investigations presented in the manuscript are based on the idea proposed by the group of V. Kruglyak [Y. Au, E. Ahmad, O. Dmytriiev, M. Dvornik, T. Davison, and V. V. Kruglyak, Appl. Phys. Lett. 100, 182404 (2012)] which was adopted later by the group of D. Grunler for the excitation of short-wavelength spin waves [24]. The switch from discs to wires enhances the excitation efficiency, here I agree with Authors, but cannot be considered as a new approach. I would recommend to publish presented results in Scientific Reports.

Few technical comments for the consideration of Authors:

- The results presented in the Fig. 1a are not fully clear. Why the frequency of magnetization precession in Co changes so strongly with changing of the magnetization orientation of YIG?
- What exactly means “SPSW efficiency”? Is it possible to get excitation efficiency out of this value (to subtract the influence of the spin-wave group velocity)?
- The Authors compare samples C1 and C2 in the Table 1. Why it is assumed that the dipolar coupling between the layers is the same? The thickness of the interlayer was increased 25 times what should also decrease dipolar coupling.

Reviewer #3 (Remarks to the Author):

I have reviewed the paper focusing on creating short-wavelength spin wave in YIG using Co nano strips. The development of exchange coupled spin waves of 50 nm length is extremely important for future efficient miniature memory/logic devices. The structure of using the Co nano strips to produce short wavelength spin waves is novel and the fact that the authors show experimental results provide reasons to continue to pursue this approach as well as potentially other approaches the author reviews in the intro section, i.e. hybrid. While the paper is well written I have a few questions below that should be addressed.

1. on line 118, could you calculate the demag factor for the Co nanowire using closed form solutions and use this calculation to compare with the value you generate from a curve fit for N_{yy} , this is more of a comment to potentially help the manuscript
2. Regarding Figure 2d, why do the experimental data points for $n=2,4,6$ begin at 500 Oe.
3. lines 170-174 -- Can authors provide an explanation of why the amplitude for mode $n=4$ is 100 times larger than CPW-excited k_1 mode
4. lines 236-240. is there any reference supporting the selection of a 10% exchange constant used in the manuscript for predicting a shift in frequency. Furthermore, could authors discuss some of

who they analytically implemented this exchange coupling between Co and YIG, i.e. this could be a reference or a description.

5. lines 267-279. The description of the significant change in transmission intensity from AP to P states is hard to understand. Could authors attempt to modify presentation to make it more clear.

6. In table 1 why are some modes missing, e.g. A3 and B1 are missing mode $n=2$.

Response to referees letter

First of all, we gratefully thank the three reviewers for taking their time to review our manuscript and providing us very detailed review comments. We found all of their comments constructive and useful, and we have taken them into account and have carefully revised the paper. The revisions have significantly improved the paper, and we hope the manuscript now meet the requirements for publication in Nature Communications. The major changes in the manuscript are highlighted in yellow. Below we provide our point-to-point responses to all of the review comments.

Remarks from Reviewer #1: The authors describe measurements on nanowires that yield short-wavelength spin waves and fast propagation velocities. These are key ingredients for any potential future implementation of spin-wave based memory-logic technology. The manuscript presents a systematic study of different thickness and structure dependence and offer routes to lowering the shortest wavelength.

Response from Authors: We greatly appreciate this very positive comment from Reviewer #1.

Remarks from Reviewer #1: The shortest wavelength is reported as 50nm in the abstract. However this is presented as an estimate in the main body of the text since the sensitivity of the measurement is not sufficient. This therefore prompts the question of whether this is an incremental advance or something more distinct. For example there are reports of wavelengths of around 100nm following a different method recently (Phys. Rev. Applied 8, 014020 (2017)).

Response from Authors: We have carefully read the very recent PRA paper (published in July 2017) mentioned by Reviewer #1, and we have found that our work differs significantly from what reported in the PRA paper, as explained below. First, the excitation methods are completely different. The work in the PRA paper was based on strain-coupled heterostructures, while our work takes advantage of periodic dynamical dipolar fields in “nanowire array”/“thin film” heterostructures. Second, spin waves in the PRA paper are confined, while in our case spin waves are propagating waves, which are very important in terms of data transfer and processing applications. Finally, in the PRA paper, the “about 100 nm” is only given in the simulations whereas the experimentally measured highest mode $n=9$ has a wavelength of about 1600 nm (considering the domain width of 7.5 μm), which is much larger than the wavelength which we measured experimentally (50 nm). To the best of our knowledge, for traveling spin waves the shortest wavelength demonstrated previously was 370 nm, as we summarized in the introduction of our manuscript. For this reason, we believe that our new approach and our demonstration, experimental and numerical, represent a significant advance from the perspective of potential spin-wave-based data transfer and computing.

We do want to acknowledge that the work reported in the PRA paper is creative and very interesting. We thank Reviewer #1 very much for letting us know about this paper and have thereby cited it briefly in our introduction (copied below) and also added it as a new reference (see Ref. 25).

“Very recently, it has been demonstrated that spin waves with $\lambda=125$ nm could be driven by the magnetic vortex states in two antiferromagnetically coupled layers²⁴; and spin waves with $\lambda=1600$ nm could be excited within magnetic domains in strain-coupled heterostructures²⁵. In both cases, however, the spin waves are confined within small elements or structures and cannot propagate over long distances.”

Remarks from Reviewer #1: Therefore the claim of shortest wavelength “ever achieved” should be toned down. Instead the importance of this manuscript is the new methodology that allows shorter wavelength and long propagation and whether this is a suitably sufficient advance. The standard dipolar exchange is certainly involved, however the authors claim that including exchange interactions is

necessary to explain the behavior and this is the key to the shorter wavelength and faster group velocity. The manuscript presents reasoned arguments and data to support this claim and is sufficiently novel and of sufficient interest.

Response from Authors: We are very happy to know that Review #1 considers our work as “new methodology”, having “reasoned arguments,” being “sufficiently novel,” and being of “sufficient interest”. Thanks! The sentence claiming “ever achieved” on page 12 has been revised: *“The corresponding wavelength of the $k=126 \mu\text{m}^{-1}$ mode is 50 nm, which is the shortest wavelength achieved so far for propagating spin waves to the best of our knowledge.”*

Remarks from Reviewer #1: In Fig. 2d the solid lines match $n=2$ and $n=4$ group velocities, however at $n=6$ there is some departure at higher field. Is this an indication of a break down of the model for higher n /field that may inhibit the path to shorter wavelengths/faster group velocity.

Response from Authors: Reviewer #1 is correct, and indeed there is a small deviation between the experimentally measured and theoretically calculated group velocities for the $n=6$ mode. Honestly, when we first saw this deviation, we wondered whether this meant the wavenumber k is so high that the spin waves are no longer in the k^2 regime but enter the $\cos(1 - kr)$ regime where the dispersion curve starts to flat out. However, we were reluctant to be so assertive on this because even the shorted wavelength (50 nm) which we measured is still much larger than the YIG lattice constant (1.2 nm). One possible explanation about the deviation is that, as the spin-wave wavelength scale gets very close to the thickness of the YIG film, there might be some influence from or coupling with the standing spin waves across the YIG film thickness. However, we are not quite sure about this at this moment, although we do plan to check this possibility using YIG films of different thicknesses in the future. We thank Reviewer #1 very much for this comment, and in response to it we have added a short discussion to the supplementary information.

Remarks from Reviewer #2: The manuscript “Long-distance propagation of short-wavelength spin waves” by C. Liu et al. represents interesting experimental studies of excitation and propagation of exchange spin waves in thin YIG films. The Authors present interesting results, nevertheless I can not recommend the manuscript for the publication in Nature Communications unfortunately. It is stated in the manuscript “This article reports a new approach for short- λ spin wave excitations that makes use of magnetization precession in periodic ferromagnetic nanowires to drive spin waves in a neighboring magnetic film.” I cannot agree with this statement. The investigations presented in the manuscript are based on the idea proposed by the group of V. Kruglyak [Y. Au, E. Ahmad, O. Dmytriiev, M. Dvornik, T. Davison, and V. V. Kruglyak, Appl. Phys. Lett. 100, 182404 (2012)] which was adopted later by the group of D. Gruninger for the excitation of short-wavelength spin waves [24]. The switch from discs to wires enhances the excitation efficiency, here I agree with Authors, but cannot be considered as a new approach. I would recommend to publish presented results in Scientific Reports.

Response from Authors: We completely understand Reviewer #2’s concern, which, we believe, arose mostly because of our insufficient discussions in our previous version, but we want to state that our approach is fundamentally different from that in the APL paper mentioned by Reviewer #2. The two major differences are as follows.

1. For the method in the APL paper the “transducer” has to work in a resonant mode. In a stark contrast, in our case the excitation of the spin waves does not require the nanowire array to be in the resonance state. As shown in Figs. 1b and 1c, the PSWSW modes appear at frequencies substantially lower than the ferromagnetic resonance (FMR) frequency of the Co nanowires. Specifically, the $n=2$ mode is more than 5 GHz away from the resonance of the Co nanowires; and in spite of such a big difference in the frequency we can still detect very nice spin-wave signals for this mode even for a propagation distance of 30 μm . Such observations cannot be explained by the “resonant microwave-to-spin-wave transducer” effect described in the APL paper.

2. The method in the APL paper cannot be used to excite short-wavelength spin waves. The reason for this is that in that method the spin waves are excited by the dipolar field from the resonant transducer that is spatially localized or confined. If we assume a pulse-like profile for the spatial distribution of the dipolar field applied by the transducer on the magnetic film, the Fourier transform yields a mostly pulse-like profile (with decaying oscillations) for the distribution of the dipolar field in the wavenumber (k) domain. As a result, the excitation efficiency is the highest for spin waves with $k \approx 0$, decreases with an increase in k , and becomes very small for high- k spin waves. In other words, this technique is suitable for the excitation of long-wavelength spin waves, but not short-wavelength spin waves. In contrast, in our approach the excitation of spin waves relies on the periodic driving (mostly through dipolar fields) by the nanowires, and the excitation efficiency is high for spin waves with $\lambda = 2a/n$ (a – array period; n – mode index), thereby enabling the efficient excitation of spin waves with short wavelengths (50 nm in the experiments and 10 nm in the simulations), as shown in Figs. 3 and S10. Thus, the wavelengths of the spin waves that can be efficiently excited are determined by different mechanisms.

Besides, in our approach, although the interlayer dipolar field is the dominate mechanism responsible for the excitation of the PSWSWs, the interlayer exchange coupling is also present and explains the frequency shifts during the switching between the P and AP states, as discussed in the third paragraph on page 13. The interlayer exchange coupling might also contribute to the “magnonic GMR” effect, as discussed in the first paragraph on page 15. In the APL paper, however, the interlayer exchange coupling is not considered at all.

Nevertheless, we appreciate this comment from Reviewer #2 and have therefore added a short discussion to page 16.

“The PSWSW efficiency is enhanced when its frequency approaches the ferromagnetic resonance (FMR) frequency of the nanowires. This is likely because, for a given microwave power, the angle of the magnetization precession in the nanowires is larger when the precession frequency is closer to the FMR frequency, giving rise to higher dynamical dipolar fields on the magnetization in the YIG in a manner similar to the resonant transducer effect reported previously⁴⁸. Note that in Ref. 48, the excitation of spin waves requires the transducer to work in an FMR state, while in this work the Co nanowires are not in resonance and the PSWSW frequencies are significant lower than the FMR frequency of the Co nanowires, as shown in Figs. 1b, 1c and S3. Note also that, unlike our approach, the method in Ref. 48 is highly efficient for the excitation of long- λ spin waves but becomes less efficient for spin waves with shorter λ .”

Remarks from Reviewer #2: The results presented in the Fig. 1a are not fully clear. Why the frequency of magnetization precession in Co changes so strongly with changing of the magnetization orientation of YIG?

Response from Authors: The frequency of the magnetization precession can be estimated by the Kittel equation $f \propto \frac{\gamma}{2\pi} \sqrt{H_0(H_0 + 4\pi M_s)}$, where H_0 is the effective external field. As one switches from the P state to the AP state, the interlayer exchange field flips the direction and thereby gives rise to a change in H_0 . Note that the strength of H_0 is much smaller than $4\pi M_s$ during the switching. Because the $4\pi M_s$ value of the Co is more than 10 times larger than that of the YIG, a change in H_0 will lead to a much larger change in f for the Co than for the YIG according to the Kittel equation.

We admit that we did not make this clear in our previous version. We thank Reviewer #2 for pointing this out, and in response to it we have added a brief discussion to page 12, which is also copied below.

“The observation that the frequency shift of the Co mode is notably larger than that of the YIG mode can be understood if we take the Kittel equation and consider the fact that the M_s value of Co is more than 10 times larger than that of YIG³⁷.”

Remarks from Reviewer #2: What exactly means “SPSW efficiency”? Is it possible to get excitation efficiency out of this value (to subtract the influence of the spin-wave group velocity)?

Response from Authors: We define the “SPSW efficiency” as the ratio of the largest amplitude of the $n=4$ short- λ spin waves measured at the detection point (antenna 2) to that of CPW-excited, long- λ spin waves.

Yes, indeed it is possible to estimate the excitation efficiency by subtracting the influence of the spin-wave group velocity. The decay length of the propagating spin wave can be estimated as (Ref. 32):

$$l_d = \frac{v_g}{2\pi\alpha f_0}$$

where v_g is the spin wave group velocity, α is the damping parameter, and f_0 is the spin wave resonance frequency. Taking device A2 as an example, the estimated “SPSW efficiency” is 106%. Considering a spin-wave propagation distance of $s = 30 \mu\text{m}$ and a decay length of $300 \mu\text{m}$ calculated for $n=4$ SPSWs, the spin wave amplitude at the excitation (antenna 1) is 1.105 times larger than at the detection (antenna 2). For the CPW-excited mode at 100 Oe, the corresponding ratio is estimated to be 1.014. With these two ratios, we can now calculate the “SPSW efficiency” at the excitation point by subtracting the influence of the spin-wave decay. The “new” efficiency is 116%, which is even slightly higher than the SPSW efficiency listed in Table 1.

It is worth noticing that here the PSWSW efficiencies shown in main text and Table 1 are relative values. It is challenging to precisely determine an absolute excitation efficiency because of the difficulty in estimating the microwave power dissipation in the CPW and the non-reciprocal propagation. The power level at the S_{11} baseline suggests a reflected power when the hybrid structure is off the resonance (R_{off}). As soon as the resonance condition is met, more power is absorbed by the spin-wave excitation of the hybrid structures and therefore less reflection power is detected by the VNA (R_{on}). To have a rough estimation, we consider the absolute excitation efficiency to be $\eta_{\text{abs}} = (R_{\text{off}} - R_{\text{on}})/R_{\text{off}}$. By this way, the absolute excitation efficiency of the CPW mode, $n=2$ PSWSW mode, $n=4$ PSWSW mode at -400 Oe can be estimated to be 5.69%, 1.57% and 5.4%. Nevertheless, such consideration ignores other power loss, such as heat dissipation and impedance mismatching, and therefore is only a rough estimation.

We believe this comment from Reviewer #2 is of interest to general readers, and we have therefore added these discussion in the supplementary information.

“If we take into account the influence of spin-wave attenuation during the propagation³², the estimated relative excitation efficiency is even slightly higher. (See supplementary information).”

Remarks from Reviewer #2: The Authors compare samples C1 and C2 in the Table 1. Why it is assumed that the dipolar coupling between the layers is the same? The thickness of the interlayer was increased 25 times what should also decrease dipolar coupling.

Response from Authors: Yes, although the dipolar coupling is “long-range” compared with the exchange coupling, a 25-nm-thick Al_2O_3 layer might be already too thick that the interlayer dipolar interaction is considerably suppressed. To address this issue, we have prepared a new sample (Sample A4) where the Al_2O_3 layer is 7 nm thick (much thinner than 25 nm). The measurements of this new sample yielded the same results – Both the $n=2$ and $n=4$ modes are clearly seen but no notable frequency shifts are observed during the switching, indicating the presence of the interlayer dipolar field and the absence of the interlayer exchange field. We appreciate this comment from Reviewer #2. We have added the new data to Table 1 and the Supplementary Information (Fig. S11) and also added a brief discussion to page 14 in the manuscript, which is also copied below.

“The same results have been observed for device A4 where a 7-nm-thick Al_2O_3 layer is inserted, although both the transmission and reflection signals are much weaker because the Al_2O_3 layer is thick (see Fig. S11). These results evidently support our conclusion that the dipolar interaction is the major

mechanism responsible for the PSWSW excitation, while the interlayer exchange coupling is responsible for the frequency shift of the PSWSWs.”

Remarks from Reviewer #3: I have reviewed the paper focusing on creating short-wavelength spin wave in YIG using Co nano strips. The development of exchange coupled spin waves of 50 nm length is extremely important for future efficient miniature memory/logic devices. The structure of using the Co nano strips to produce short wavelength spin waves is novel and the fact that the authors show experimental results provide reasons to continue to pursue this approach as well as potentially other approaches the author reviews in the intro section, i.e. hybrid. While the paper is well written I have a few questions below that should be addressed.

Response from Authors: We are very glad to know Reviewer #3 considers our work “extremely important” and “novel” and our paper “well written”.

Remarks from Reviewer #3: 1. on line 118, could you calculate the demag factor for the Co nanowire using closed form solutions and use this calculation to compare with the value you generate from a curve fit for N_{yy} , this is more of a comment to potentially help the manuscript.

Response from Authors: This is a very helpful suggestion. In fact, we feel very lucky that Reviewer #3 brought up the estimation of the demagnetization factor, which made us revisit our calculation and realize that we actually had mixed up the xyz axis definition between ours and that in Ref. 34 (Ding, J., Kostylev, M. & Adeyeye, A. O. Phys. Rev. B 84, 054425 (2011)). Their N_{yy} is actually our N_{xx} . Therefore, in our case what really matters is the N_{xx} , while N_{yy} is approximately zero due to the large aspect ratio. The N_{xx} value calculated using the dimensions of the Co nanowires is about 0.23, which is notably larger than the value from the fitting (0.03). A similar significant disagreement also occurs in the work by Ding *et al.* reported in Ref. 34, and it can be understood if we consider the effective dipole pinning at the edges of the wires (not considered in the calculation). In the new version, we have corrected the xyz definition and have also added a brief discussion to page 6, which is also copied below.

*“The calculation based on closed form solutions and the Co nanowire dimensions yields $N_{xx} = 0.23$, which is significantly larger than the fitting value. This disagreement was also found by Ding *et al.*³⁴ and was explained as the influence of the effective dipole pinning of dynamic magnetization at the element edges.”*

Remarks from Reviewer #3: 2. Regarding Figure 2d, why do the experimental data points for $n=2,4,6$ begin at 500 Oe.

Response from Authors: That was because we did fine measurements for a field range of 500-800 Oe only, as the fine measurements are time consuming. We believe this question is of interest to general readers, so we have carried out additional measurements at low fields and have updated Fig. 2d with the new group velocity data points. The new figure shows the data for a field range of 0-800 Oe.

Remarks from Reviewer #3: 3. lines 170-174 -- Can authors provide an explanation of why the amplitude for mode $n=4$ is 100 times larger than CPW-excited k_1 mode

Response from Authors: The efficiency of the PSWSW excitation is very high, and the scale of "100 times" was extracted from the experimental results. The reason for this is that the Co nanowires provide very strong, spatially periodic dipolar fields to force the magnetization in the YIG to precess in a certain periodic manner, leading to the excitation of spin waves with certain wavelengths associated with the period of the nanowire array. In a stark contrast, the excitation using the conventional CPW technique relies on the CPW-produced microwave magnetic field that, unlike the dipolar fields from the nanowires, has a relatively uniform spatial distribution and therefore can efficiently excite long-wavelength spin waves, but not short-wavelength waves. Note that the dimensions of the CPW structure

are much larger than those of the nanowires. In line with this comment from Reviewer #3, we have added a short discussion in the supplementary information as below.

“These results clearly indicate that the periodical driving force from the nanowires can efficiently excite PSWSWs, whereas the microwave magnetic field from the CPW is relatively uniform and therefore cannot efficiently excite short-wavelength spin waves.”

Remarks from Reviewer #3: 4. lines 236-240. is there any reference supporting the selection of a 10% exchange constant used in the manuscript for predicting a shift in frequency. Furthermore, could authors discuss some of who they analytically implemented this exchange coupling between Co and YIG, i.e. this could be a reference or a description.

Response from Authors: We thank Reviewer #3 for this suggestion. We searched the literature and failed to find references discussing about the exchange coupling between Co and YIG (especially with a thin nonmagnetic spacer in between them). However, we did find a useful reference: "Hybrid yttrium iron garnet-ferromagnet structures for spin-wave devices" A. Papp, W. Porod, and G. Csaba, *J. Appl. Phys.* **117**, 17E101 (2015), where they took $A_{\text{interlayer}} = 0.5 \times 10^{-12} \text{J/m}$ between Permalloy and YIG. This value is actually very close to that we used in simulations ($4.875 \times 10^{-12} \text{J/m}$). In our revised version, we have added this JAP paper as a new reference (Ref. 42). Honestly, we do not precisely know how strong the interlayer exchange is, so we tried a value which is 10% of the exchange constant in the YIG and happened to be very close to the value in that JAP paper. The simulation results show reasonable agreements with the experimental results. More systematic simulation studies considering different exchange coupling at the interface have been carried out recently and will be presented in detail in a separate report.

Remarks from Reviewer #3: 5. lines 267-279. The description of the significant change in transmission intensity from AP to P states is hard to understand. Could authors attempt to modify presentation to make it more clear.

Response from Authors: We thank Reviewer #3 for this suggestion. In line with it, we have added some discussions to page 16, which are also copied below.

“Such a critical dependence of the transmission signal on the magnetic configuration might be due to three possible reasons. (1) The magnetic moments in the YIG precess in the same manner as that in the Co for the P state but in an opposite manner for the AP state, resulting in a different efficiency for the Co-drive-YIG scenario. (2) The spin current generated by the Co magnetization precession exerts anti-damping-like and damping-like torques^{46,47} on the YIG magnetization for the P and AP states, respectively, giving rise to a larger amplitude for the P state than the AP state. (3) It is possible that the interlayer exchange coupling being FM-like helps the PSWSW excitation in the P state.”

Remarks from Reviewer #3: 6. In table 1 why are some modes missing, e.g. A3 and B1 are missing mode $n=2$.

Response from Authors: Yes, some PSWSW modes are very strong, while other “expected” modes are too weak to be measured. This is because the actual intensity of a particular PSWSW mode depends on the interplays of several different factors, including the following three.

1. **Excitation efficiency.** The excitation efficiency can be different for different PSWSW modes. For example, the excitation efficiency is stronger for the modes with frequencies close to the FMR frequency of the nanowires than for those with frequencies very different from the nanowire FMR frequency. For sample B1, for example, the $n=6$ and $n=8$ modes are closer to the Ni nanowire FMR mode and therefore show sizable signals; and the $n=4$ and $n=10$ modes are weaker but still detectable with reasonable signal strength. However, the $n=2$ and $n=12$ mode can hardly be detected

because their frequencies are far different from the nanowire FMR frequency. This is the reason why only $n=4,6,8,10$ modes are included in Table 1.

2. Decay length. The spin-wave decay length is strongly dependant on the frequency as shown in the newly added Eq. S2,

$$l_d = \frac{v_g}{2\pi\alpha f_0}.$$

This suggests that for a given propagation distance, the higher the frequency is, the shorter the decay length becomes and the weaker the transmission signal is.

3. Propagation distance. The detection distance plays a critical role in the experiment because what we are measuring is the transmission signal of the PSWSWs. For example, sample A3 has the longest detection distance, i.e. the longest separation between the two integrated CPW antennas, and the transmission is expected to be weaker than in other samples. As a result, only the mode with the highest excitation efficiency and reasonably long decay length can be detected, which is the $n=4$ mode in this case.

We thank Reviewer #3 for inspiring the above discussions. Considering the length limitation of the text, we now added these discussions in the Supplementary Information (after Eq. S2) and also added one sentence referring to those discussions in the text. This sentence is on page 17 and is also copied below.

“.....More discussions on why some modes are not detectable are given in the Supplementary Information.....”

Finally, we hope the revised manuscript can now meet the requirements for publication in Nature communications.

With our kind regards,

Haiming Yu and Mingzhong Wu (on behalf of the authors)

Reviewers' comments:

Reviewer #1 (Remarks to the Author):

The authors have suitably addressed the previously stated concerns in this re-submission. There are no major changes to the manuscript and as such no further alterations are suggested.

Reviewer #2 (Remarks to the Author):

The Authors have answered my scientific questions and have argued convincingly about the novelty of their research. I accept their arguments and recommend the manuscript for publications in its current form.

Reviewer #3 (Remarks to the Author):

the authors have adequately responded to my queries

REVIEWERS' COMMENTS:

Reviewer #1 (Remarks to the Author):

The authors have suitably addressed the previously stated concerns in this re-submission. There are no major changes to the manuscript and as such no further alterations are suggested.

We thank Reviewer #1 for confirming that no major changes are needed.

Reviewer #2 (Remarks to the Author):

The Authors have answered my scientific questions and have argued convincingly about the novelty of their research. I accept their arguments and recommend the manuscript for publications in its current form.

We thank Reviewer #2 for his acceptance of our arguments and recommend publications of our manuscript.

Reviewer #3 (Remarks to the Author):

The authors have adequately responded to my queries

We thank Reviewer #3 for his positive confirmation.